# Why People Settle in Shrinking Communities: A Unified Framework of Psychological, Environmental, and Cognitive Factors

**Siyi An** [1,*] , **Toshiaki Aoki** [1] and **Atsushi Suzuki** [2]

[1] Graduate School of International Cultural Studies, Tohoku University, Sendai 980-8576, Japan; toshiaki.aoki.a1@tohoku.ac.jp

[2] Department of Civil Engineering, Meijo University, Nagoya 468-8502, Japan; atsuzuki@meijo-u.ac.jp

* Correspondence: an.siyi.p3@dc.tohoku.ac.jp

**Abstract:** The interpretation of settlement behavior in shrinking areas can provide insights into sustainability strategies in shrinking communities. However, the choice to settle in areas undergoing shrinkage is hard to interpret when considering residents as rational decisionmakers. To attain a deeper understanding of this decision-making process, a framework of residential decision making (RDM) considering a subjective environmental evaluation dimension, psychological dimension, and cognitive dimension is proposed. This process was further validated by conducting a questionnaire survey in Japanese communities. The results of the structural equation modeling reveal that the RDM framework proposed is applicable to RDM in shrinking communities. By considering geographical differences, we further found that residents in suburban communities tend to consider overall satisfaction with their location when deciding whether to stay, whereas residents in mountainous communities value emotional satisfaction factors such as place attachment when considering continuous residence. Different residential preferences contributing to the formation of RDM factors were also revealed between communities. The results of this study imply that sustainable development strategies to assist shrinking communities should be tailored to their geographical characteristics. Further, a regional design that can enrich residential experiences and neighborhood communication is important for promoting population settlement.

**Keywords:** residential decision making; residential environment; immobility; shrinking communities

## 1. Introduction

With long-term population decline and aging occurring globally, region shrinkage has become an accepted reality for regional planners in recent years [1]. To counteract this, measures such as promoting citizen participation and improving quality of life (QoL) are advocated as appropriate policy responses [1–3]. However, there is no concrete methodology for realizing the goal of preventing region shrinkage [2]. In this regard, the understanding of residential decision making (RDM), a cognitive process concerning how people select their places of residence, may provide a meaningful contribution to defining strategies for the sustainable development of shrinking communities [4].

Since population decline is usually considered as a critical manifestation of the shrinkage phenomenon [2], studies on the shrinkage context usually involve the topic of RDM. Studies on RDM have been conducted in European countries such as Portugal [4,5] and Germany [6], and in Asian countries such as Japan [7,8] and China [9]. These studies have attempted to understand RDM in shrinking cities by capturing the effects of QoL (quality of life) [9–11], satisfaction with living [5,6], and pull or push factors for staying in the current place [4]. Their results generally point out that a city experiencing shrinkage is not necessarily accompanied by a decline in QoL [9–11]. The subjective well-being may be well-maintained [6] and the QoL even increase in shrinking cities [10,11]. Furthermore,

environmental factors such as safety [5,10], living conditions [5], transportation and accessibility [7,10], shrinking atmosphere [4,5], and social ties [4,5] were reported as factors influencing staying intention. Economic factors, such as employment conditions [4,5], were reported to stimulate residents to leave shrinking cities. However, in addition to these environmental variables, some emotional factors (e.g., place attachment [5,7]) and even cognitive bias (e.g., residential satisfaction is found to be hard to improve by long-term residence [5]) were also found to influence the RDM. These findings contradict the traditional assumption that socioeconomic factors or place utility explain RDM [12]. It implies that focusing on residential improvements alone may not be sufficient to advance population settlement policies in shrinking cities. The psychological impact is also of concern. Considering that the shrinkage phenomenon is fast becoming an unavoidable issue for the future of regional planning, it is necessary to expand the understanding of RDM and develop a comprehensive framework to understand RDM in shrinking communities. This investigation can aid in filling the existing theoretical gap in location choice theories explained by rational choice, and provide political implications for policy makers to define strategies for stabilizing and slowing the decline of regional populations in the context of shrinkage [13].

In studies on shrinkage, RDM is often understood in keeping with the spatial scale of cities. The term "shrinking city", which is defined as "a densely populated urban area with a minimum population of 10,000 residents that has faced population losses in large parts for more than two years and is undergoing economic transformations with some symptoms of a structural crisis" [14] is used in a large number of studies e.g., [1,2,4–6]. However, given that a city may contain both areas of population growth and population decline [15] and the unclear relation between QoL and tendency of urban shrinkage [9–11], studies based on the urban scale to understanding RDM may not explain the relationship between settlement behavior and their living environment effectively. In contrast, shrinkage at the neighborhood scale is more indicative of a potential feedback loop between population decline and the consequences of shrinkage (e.g., vacant housing, underutilized infrastructure, and reduced tax revenues). Neighborhood-level shrinkage seems more likely to reflect a dynamic shrinkage process of a geographic area from population contraction, decay or even extinction. Based on this, it can be argued that understanding the RDM of residents based on a grasp of the geographic context within a community is vital in constructing specific measures to cope with shrinkage.

In this study, communities undergoing shrinkage in Japan were considered case areas. As a country undergoing local shrinkage caused primarily by an aging population and decreasing fertility rate, the Japanese government has enacted a series of laws since the mid-1960s to mitigate this trend [16]. Between 1995 and 2015, approximately 72% of Japanese municipalities experienced population decline [17]. By 2020, Japan's elderly population represented >50% of the total population in 29.2% of Japanese communities [18]. On the basis of this trend, shrinkage in Japan has been considered as a future norm [2]. Differently from the international research with a viewpoint based on a city scale, studies on shrinkage in Japan also attempt to understand shrinkage on the neighborhood scale [2]. A common demographic indicator, ≥50% of the total population aged ≥65 years, is commonly used to identity whether aneighborhood is experiencing shrinkage [8,19]. According to a 2020 survey on the current status of Japanese districts from the Ministry of Land, Infrastructure, Transport and Tourism, 75.3% of districts with an elderly population of ≥50% have difficulty maintaining community functionality [18]. Based on this, we assume a neighborhood where ≥50% of the total population aged ≥65 years tend to undergo shrinkage and term them as the shrinking community in the present study.

Based on the aforementioned considerations, this study aims to propose an RDM model and apply it to explain settlement behavior in shrinking communities. Residents in shrinking communities in Japan show a strong intention of continuous residence in their current location, e.g., [7,8]. The phenomenon of settling trends may particularly help urban

planners to better comprehend the pull factors of shrinking communities. Investigations in Japan may provide worthwhile findings for regions facing similar issues around the world.

### 1.1. Previous Research on RDM

RDM is a topic with a long history of multidisciplinary contributions from fields such as urban planning, geography, sociology, economics, cultural anthropology, and psychology [12]. For this reason, RDM (linked to keywords such as *residential mobility* or *residential location choice*) often appears in research from a variety of academic fields. Despite this range of research approaches, all existing studies converge on a single question: how do humans select their place of residence?

Research on RDM can be segregated into macro-level, meso-level, and micro-level on the basis of the previous classification [12,20]. Studies on the macro-level are concerned with explaining aggregate migration behavior based on the assumption of neoclassical theory that individuals are motivated by maximizing income. Utilizing wage differential to interpret residential mobility is a common viewpoint [12]. Studies on the meso-level are concerned with structural opportunities and constraints such as kinship, neighborhood, and ethnicity, which form social contexts of choice [20]. Studies on the micro-level attempt to interpret individual residential mobility behavior within the context of psychological decision-making processes [20]. The concept of an "intentionally rational decisionmaker" has become a dominant assumption in RDM [21].

For the sake of interpreting settlement behavior, the present study focuses on the understanding of RDM from a behavioral perspective. From this viewpoint, RDM is considered a response to the place utility associated with an individual's place of residence. Wolpert proposes that residents can rationally calculate the net utility gleaned from their residential environments or social interactions [21]. The evaluated utility is compared with the subjective environmental stress residents experience in their living place, the latter of which functions as a trigger for moving behavior. Speare [22] furthermore argues that residential satisfaction can be used as a measurable intervention variable to explain the influence of residential environment on intention of residence. A body of studies based on the model of residential satisfaction demonstrates that residential satisfaction enhances settlement intention, whereas dissatisfaction enhances motivations for relocation behavior [23–26]. Offering a comprehensive assessment of spatial, human, and functional aspects of place, the residential satisfaction model has become one of the most widely adopted methods for predicting residential intention and evaluating QoL [27].

However, some studies point out the limitations of residential satisfaction in interpreting mobility behavior. For example, the constraints perspective rejects the assumption of utility maximization and argues that RDM considers external constraints such as cost of dwelling and travel time to work [28]. Additionally, the housing market, ownership [29], political barriers [30], and life stage [31] are strong social structural constraints that directly impact willingness to move. For this reason, it can be argued that pooled utility can only partially explain the rationality of the entire RDM process. Actual residential mobility is likely to exhibit more limitations [32,33].

In addition, the effects of internal criteria such as self-selection [34], lifestyle [35], and residential preference [35,36] also influence the process of RDM, particularly in recent decades. It is generally argued that the greater the fit between these factors and a particular environment, the more likely an individual will be to feel satisfied or desire to move to this location [37,38]. Conversely, a poor fit will enhance relocation intention [39]. For example, a preference for short-term travel can motivate residents to live in a convenient urban region [40], and matching between travel preference and location features may further enhance residential satisfaction [41]. This indicates that a person's choice of residence may not be based on environmental factors but rather on internal criteria. Here, the propensity toward seeking cognitive harmony becomes the biggest motivation in RDM.

In sum, a general mechanism of RDM could be assumed to underpin all perspectives: a resident is a rational decision maker who seeks the satisfaction of both their external and internal needs by residential conditions in consideration of current constraints.

### 1.2. RDM in Shrinking Communities

Research on region shrinkage began during the last century; however, it was predominantly concerned with a macro-scale understanding of the shrinking phenomenon focusing on, for example, the interactions of buying and selling houses [42], or the change in land consumption as shrinkage progresses [43,44]. Studies referring to individual RDM in shrinking cities have only emerged in recent years [2].

The most representative research focuses on shrinking cities in Portugal. Using a face-to-face questionnaire survey conducted in four shrinking Portuguese cities, Guimarães et al. [4] found that residential environment factors and economic activity are the determinant factors in RDM in these cities. Additionally, in encouraging continuous residence, psychological factors like social ties and place attachment were important. Conversely, shrinking atmosphere or surroundings were interpreted as push factors for leaving. Barreira et al. [5] furthermore reported *accessibility* (which overrides *living conditions*), *recreational and environmental amenities*, and *social ties* as common factors that contribute to residential satisfaction levels in shrinking communities. Other studies focus on QoL in the context of shrinkage, which may be important in understanding RDM. By comparing the objective QoL and subjective QoL of residents in shrinking cities between 2013 and 2017, Liu et al. [10] found that while subjective satisfaction tended to decrease in rapidly shrinking cities, it slightly increased in slowly shrinking cities. A study in Germany, however, indicates that subjective satisfaction depends on the geographic area in which people live rather than on the area's rate of shrinking [6]. These studies suggest that a shrinking phenomenon is not necessarily linked to a decline in residential satisfaction or QoL; therefore, the correlation between residential environment and RDM is still not entirely clear.

Comparing the aforementioned results with existing residence choice theories, it is revealed that there are some gaps between research and theory that need to be bridged. For example, existing RDM theories exhibit limitations in interpreting staying behavior [45]. As Schewel [45] points out, although continuous residence behaviors have historically been viewed as the absence of RDM, they are indeed the result of decision-making. Factors involved in this decision-making process include "retain" factors, which cause people to view their places of residence as preferable to other places, "repel" factors, which diminish the desire to progress in RDM, and "internal constraints" such as risk attitudes. Clearly, the complexity of continuous residence behavior should be given attention. When considering continuous residence as an important policy goal for sustainable development in shrinking communities [13], it is important to fully understand the RDM process that leads to continuous residence.

In addition, noneconomic factors such as place attachment have also been shown to play a notable role in empirical studies of shrinking communities [5,7,9]. This suggests that the existing notion that considers socioeconomic and environmental factors as the rational criteria underpinning RDM presents limitations in explaining RDM in shrinking communities. Therefore, an empirical approach to RDM that considers a broader definition of rationality is required.

Another important point to note here is that residents' understanding of residential environment and QoL differs depending on region [4,9]. However, few studies consider the possibility that RDM could change depending on region or residential environment. This topic requires further investigation.

On the basis of these considerations, the purpose of this study is described in the following two points:

- To propose a comprehensive RDM framework considering psychological and cognitive factors to interpret settlement behavior in shrinking communities in Japan

- To determine whether RDM differs between communities with different geographical features

## 2. Theoretical Framework

### 2.1. An RDM Structure

Through the development of social sciences and neuroscience, the role of emotions and cognition in decision making has been increasingly demonstrated [46]. However, these dimensions are still often overlooked in RDM studies because they tend to be interpreted as "uneconomic" or "irrational" [45]. Although these variables have been given attention in some studies e.g., [47–52], the overall influence they have during the RDM process remains difficult to assess, making it challenging to adequately explain some moving behaviors such as the choice to stay in a shrinking community. In order to deepen the understanding of rationality in the RDM process, it is crucial to establish a multidimensional framework for assessing RDM.

In this study, an RDM framework consisting of multidimensional factors was assumed, as displayed in Figure 1. It is a two-layer structure presuming that RDM is directly influenced by RDM factors and indirectly influenced by geographical elements in one's region of residence. The RDM factors are divided into three dimensions: subjective environmental evaluation dimension, psychological emotion dimension, and cognitive intervention dimension. The subjective environmental evaluation dimension refers to the collective functional utility attained from the residential environment; the psychological emotion dimension refers to the overall emotional satisfaction gained from living in a specific place; and the cognitive intervention dimension refers to the variables that influence RDM at the cognitive level.

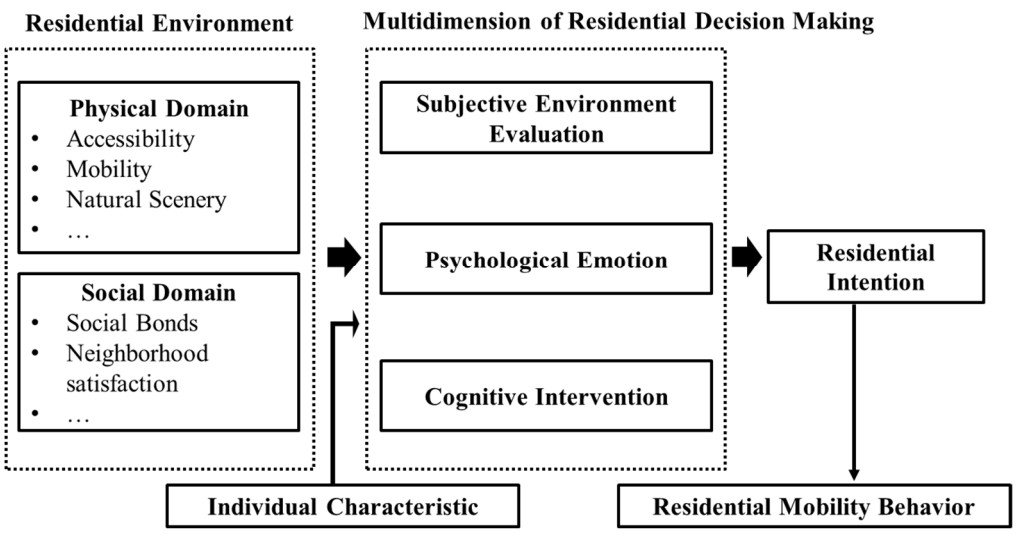

**Figure 1.** Framework of Residential Decision Making.

The influence of environmental factors on RDM is achieved through the intervention of RDM factors. For example, convenient accessibility on its own does not necessarily motivate residents to settle; however, residents might feel motivated to stay by the residential satisfaction it brings. Likewise, social bonds and the beauty of the natural environment can contribute to staying intention through facilitating place attachment. Thus, environmental elements around residence function like an environmental resource to form each RDM factor, affecting RDM. Furthermore, the weighing of environmental factors would be moderated by the different residential needs shown by residents.

The representative RDM factor in this study is residential satisfaction [26]. This dimension, although often used in the interpretation of RDM, is intertwined with a great deal of subjective cognitive processing. As the process of how internal criteria influence residential intention shows, higher residential satisfaction may derive from individual

residential preferences or lifestyles [35,37,38]. For example, even though regional amenities are rated differently, a similar level of satisfaction could be attained depending on personal preferences [53]. Conversely, residential satisfaction has also shown limitations in the decision-making process. The phenomenon of "bounded rationality" proposed by Simon [54] states that because of limitations on information and cognitive resources, a decisionmaker may look for a satisfactory alternative rather than choosing the option with maximum utility. This limitation manifests in RDM because once a resident feels satisfied with their place of residence, their moving intention decreases despite the possibility that they could find deeper satisfaction elsewhere [22]. Thus, although subjective environmental evaluation is a dimension reflecting objective living conditions, it should also be differentiated from them.

Place attachment [23,47,49], neighborhood reputation [48,55], etc., can all be included in the dimension of psychological emotion. The most frequently explored factor of this dimension is place attachment. Although many studies have shown that place attachment can enhance staying intention, most of these studies tend to attribute its impact to the function of social ties such as identification with roots [56], embeddedness in the local area [57], and connection with family [49]. The emotional function of place attachment has not been paid due attention. Scannell and Gifford [58] state that place attachment is also an affective result of the process of interacting with a place. Scannell and Gifford [59] reported that when people recall the image of their attached place, positive feelings such as relaxation, belonging, comfort, and security could be aroused. They further demonstrated that the visualization of an attached place can facilitate subjective well-being [60]. Ratcliffe and Korpela [61] found that the recollection of positive memories associated with an attached place can evoke restorative potential. Therefore, it can be argued that place attachment can provide considerable emotional value for a resident if they feel attached to their place of residence. Bandyopadhyay et al. [62] noted two ways in which emotions influence decision making: one is functionally similar to place utility, in which people attempt to maximize positive emotion; the other stresses how final decisionmaking behavior is affected by emotions experienced at the time of the decision. These processes can help explain the effects of emotional factors on RDM.

The cognitive intervention dimension encompasses the cognitive variables that may impact RDM. Factors in this dimension can be subdivided into two levels: social and individual. The social level includes cognitive interventions arising from the internalization of cultural and social contexts. This includes social expectations from a local social network [63], migration intentions of other family members [50], and obligations to family members or property [47], all of which were found to significantly affect RDM. Such cognitive interventions represent "thick rationality" generated by family, group, and cultural contexts [64]. Conversely, factors from the individual level refer to individual cognitive processes or experiences. Jaeger et al. [52] finds that individuals who are likely to take risks are more willing to migrate. Feijten et al. [51] reveals that individual residential experience can lead to different migration tendencies. It could further be argued that the effect of these factors on RDM processes is relatively unconscious to decisionmakers. In summary, although a certain amount of research exists on the cognitive intervention dimension, it remains a marginal lens in RDM research overall [45].

Residential environment contributes to RDM indirectly by forming RDM factors in this model. Both physical and social environmental factors can be seen as environmental resources. For instance, physical elements such as convenience of neighborhood facilities [65] and function of dwelling [55] are related to the formation of residential satisfaction. Natural environment, architecture, etc., are linked to the formation of place attachment [58,66]. Social elements such as social bonds, social capital, and interaction within the neighborhood further contribute to both residential satisfaction and place attachment, e.g., [22,55,66,67]. In the dimension of cognitive intervention, factors are more likely to be influenced by environmental factors from the social domain. Berry notes that migration can come with a process of "cultural shedding" and "cultural learning" in order to integrate into a new

social context. This eventually manifests itself in changes in diet, the way people dress, and even values [68].

Individual attributes such as income, family structure, and age are assumed to affect RDM indirectly. For example, people with higher incomes can achieve mobility intentions in the short term, whereas the moving behaviors of people with lower incomes are more vulnerable to housing policies [23]. Such attributes affect residential mobility through a set of common needs or constraints [28,69].

### 2.2. Hypotheses

Following the theoretical framework of RDM outlined in the previous section, the RDM structure in shrinking communities can be represented as shown in Figure 2. The hypotheses of this study focus on validating the effects of the multidimensional RDM factors on residential intention.

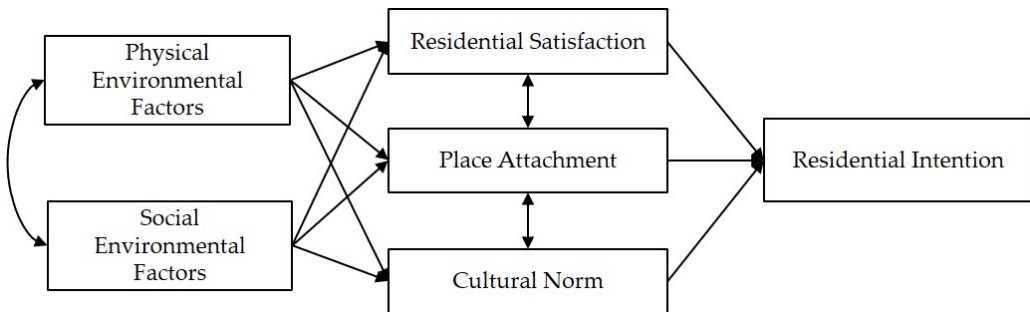

**Figure 2.** Residential Decision Making Structure in Shrinking Communities.

In the subjective environmental evaluation dimension, special attention is given to the effect of residential satisfaction on residential intention. Previous results show that even in shrinking cities, residential satisfaction remains an important factor in strengthening staying intention [5]. The effects of residential satisfaction can also be assumed to exist in Japanese shrinking communities.

**Hypothesis 1 (H1).** *It can further be assumed that the more satisfied a person feels with their residence, the stronger their desire and intention will be to continue living there.*

Moreover, the environmental factors forming residential satisfaction are interpreted as physical factors such as transportation, accessibility [10,70], safety [10,70], and social factors such as social trust [70]. These were reported as elements of QoL in shrinking cities.

Place attachment, the representative factor of the psychological emotion dimension, has been demonstrated to strongly promote intention of continuous residence in regions without a favorable residential environment, such as locales experiencing shrinkage, extreme weather, or terrorist attack [4,46,71].

**Hypothesis 2 (H2).** *Hence, it can be assumed that the more attached a resident feels to their place, the stronger their desire will be to continue living there.*

Although there is no specific research on how place attachment develops in the context of shrinkage, available relevant research suggests that it is derived from environmental factors such as natural assets and social capital [58]. Thus, in this study, the effect of environmental factors from both the physical and social domains is considered in the formation of place attachment.

In considering the cognitive intervention dimension, the effect of cultural norms on residential intention is the main factor. Obligation to family members, property, or assets [47], protecting ancestral land, and inheriting a family estate [72] have been reported among the top reasons for residents to settle in areas with an unfavorable residential environment. In Japan, this trend is deeply rooted in the historical culture of the "family

system" and affects residential intention through the cognitive constraint of familial and cultural obligation.

**Hypothesis 3 (H3).** *Hence, it can be assumed that strong adherence to cultural norms is positively correlated with the tendency for residents to continue living in a current location.*

It is worth noting here that the formation of cultural norms is dependent on residential environment. Because shrinking communities have more elderly residents and lower population mobility, the local cultural context can be expected to be more conservative and homogeneous. This social environment makes it easier for residents to develop strong cultural norms. On the basis of the aforementioned considerations, it can be assumed that both physical and social environmental factors are closely related to cultural norms that affect RDM.

As a final note on framework, RDM predictors from the described dimensions might weigh differently in staying intention. A study conducted in depopulated regions in Japan shows that residents in central areas continue to live there because of convenience, whereas those in peripheral areas settle because of their attachment to place [7]. We speculate that this may be because shrinking communities often lack social vitality and infrastructure [73]; thus, desirable levels of residential satisfaction could be difficult to maintain. Rather, the psychological satisfaction attained from place attachment can be seen as the determining factor in settling behavior.

**Hypothesis 4 (H4).** *Thus, it can be assumed that place attachment plays a greater role in settling intention in shrinking communities than residential satisfaction.*

### 3. Research Design and Methods

Based on the research purpose and hypotheses mentioned above, a cross-sectional questionnaire survey was designed to answer the question of why residents settle in shrinking communities. Furthermore, the different effects on RDM across different types of communities were considered.

Both quantitative and qualitative research methods were used in order to understand the intention of residents to staying in their current location. An example of a quantitative research method is the study of Guimarães et al. [4], in which pull and push factors for people to stay in shrinking cities were abstracted based on items composed of environmental factors (e.g., climate, shrinkage landscape, accessibility for daily facilities, etc.), social networking (e.g., sense of community, etc.) and socioeconomic factors (e.g., work, house rent, etc.). A qualitative study was conducted by Adams [47] in which the reasons for both staying and moving were asked in regions whose populations experience negative health and livelihood impacts from climate-related phenomena, carried out via interview survey. Each of these two types of methods has strengths in exploring the relationship between the proposed variables and settlement behavior, as well as in understanding residents' motivations for staying. In order to provide a contribution to policymaking in shrinking communities, and based on insights from the residents' point of view, we combined both research approaches in our questionnaire. We consider that an RDM theoretical framework in shrinking communities can be better understood and improved on the basis of capturing the residents' reasons and motivations for staying.

#### 3.1. Survey Areas

Based on the above considerations, we selected a Japanese Prefecture as our study area, Miyagi prefecture, which is undergoing population decline. Miyagi prefecture is located the northeast of Japan, with a population of approximately 2.30 million in 2021 [74]. It is the economic and transportation center for the Tohoku region. We chose Miyagi prefecture for two reasons. First, the regional shrinkage has been pronounced in this region and is expected to advance in the future. According to the 2019 population census for Miyagi prefecture, a population decrease was observed in 66% of the mesh (1 km$^2$) from

2010 to 2015. In addition, 60% of the mesh is expected to have its population halved by 2050 [75]. Thus, region shrinking is a major issue in Miyagi prefecture. The second reason for choosing this region is that Miyagi prefecture represents multiple land use types. According to the status of land use for Miyagi prefecture in 2018, 11.2% of the total districts are urban and 17.4% are farmland [76]. This implies that shrinking communities in Miyagi prefecture could cover comprehensive land use types.

We selected survey areas based on the 2015 Japanese population census. Japanese population census aims at obtaining basic data on the actual conditions of people and households, which may then be applied in policymaking. It covers the statistics of gender, age, population migration, etc., [77] in Japan. We used statistical data based on small areas [78]. In this data, Japan was divided into 4342 small areas (including uninhabited areas) by neighborhoods and clear geographical signs such as roads, rivers, railways, and waterways. The average population for each area was 476.2, and the average number of households was 191.8.

Based on the definition of shrinking community, 186 small areas from Miyagi prefecture where ≥50% of the total area population is aged ≥65 years were selected as primary survey areas. To ensure that inhabitants of these areas could make residential decisions based on their own will, areas located in municipalities or cities impacted by the Great East Japan Earthquake were excluded. The RDM of residents in such areas is expected to have been influenced by policy guidance. Areas equipped with facilities for elderly people were also excluded as these facilities may increase the proportion of elderly people residing in these areas. Ultimately, thirty-nine small areas were selected as survey targets.

### 3.2. Classification of Shrinking Communities

Hierarchical cluster analysis was conducted using Ward's method to cluster the characteristics of the selected areas before data collection. We used the standardized value of six variables, representing geographical characteristics and population characteristics, in the analysis. Geographical characteristics of shrinking communities include whether DID (densely inhabited districts) areas are included, altitude, distance to the nearest station, and distance to a town hall. Population characteristics comprise the ratio of elderly population and ratio of fluid population. Altitude was collected from the GSI (Geospatial Information Authority of Japan) [79]. Distance to the nearest station and distance to a town hall were obtained from Google Maps by searching the route for driving. The ratio of elderly population, ratio of fluid population, and the presence/absence of DID area data was obtained from the Japanese population census for small areas. The cluster analysis results show that the percentage change of heterogeneity for the final step (68.4%) was greater than the previous step (12.4%). Thus, we adopted two clusters as the final cluster solution. Table 1 shows the mean of the variables for each cluster.

Communities in Cluster 1 are located at lower altitudes and at closer distances to the nearest train stations and town halls. Some of them have densely inhabited district areas with larger mobile populations and lower elderly populations. Therefore, these communities are likely located in suburban flat terrain areas, which have convenient accessibility to local central areas and stations. Thus, we have termed Cluster 1 "Suburban Communities". Communities in Cluster 2 are located at higher altitudes and 20 km away from the nearest stations and town halls, on average. None of them have DID areas. Communities in this cluster have smaller mobile populations and larger elderly populations. These communities are likely located in mountainous areas where access to local central areas or public transportation is generally inconvenient. Thus, we have termed Cluster 2 "Mountainous Communities".

**Table 1.** The Results of Cluster Analysis.

| Variable | Cluster 1 | Cluster 2 |
|---|---|---|
| | (N = 20) | (N = 19) |
| **Population Characteristics** | | |
| Ratio of Elderly Population | 54% | 58.37% |
| Ratio of Mobile Population [1] | 12.40% | 6.62% |
| **Geographical Location Characteristics** | | |
| With or Without DID Areas | 25% | 0% |
| Altitude (m) | 36.2 | 255.2 |
| Distance to the Nearest Station (km) | 2.3 | 15.1 |
| Distance to the Central Government Office (km) | 5.4 | 20.7 |

Notes: In this study, Clusters 1 and 2 are named as suburban and mountainous communities, respectively. [1] Ratio of Mobile Population was calculated from the percent of population whose residential length is within 5 years. DID (Densely Inhabited District).

*3.3. Questionnaire*

The questionnaire included the following: items concerning three RDM predictors (residential satisfaction, place attachment, and cultural norms); an assessment of residential environment; questions regarding residential intention; and individual attributes such as gender, age, and homeownership.

Residential satisfaction was measured using two options: "I am satisfied with the overall living environment in this area", or "I am satisfied with my life in this area". Place attachment was measured using eight items modified from Williams and Vaske [80], including place identity (affective bonds) and place dependence (instrumental bonds). Cultural norms were measured using three items aimed at grasping the extent to which residents think they should obey the traditional obligation of protecting the ancestral tomb and inheriting family property. All of the aforementioned 13 items were measured using a 5-point Likert scale ranging from 1 (strongly disagree) to 5 (strongly agree).

Assessment of residential environment consisted of 28 items. Environmental factors that could contribute to the formation of residential satisfaction and place attachment were measured. The measurement of physical environmental factors considered accessibility to public facilities, public transport, mobility, and sense of security, factors contributing to QoL in shrinking Japanese communities [8]. The attractiveness of local assets was also considered, which is strongly associated with place attachment.

Concerning social environmental factors, social capital was measured. Although both social ties and social capital can contribute to the formation of residential satisfaction and place attachment, social capital can better represent the value such connections hold [81]. Thus, it can be supposed that the measurement of social capital could explicitly represent the output of social interaction. Thus, nine modified items [67] were used to measure reciprocity, trust, and community activity. Overall, 28 items were used in the assessment of residential environment in shrinking communities. A 5-point Likert scale ranging from 1 (strongly disagree) to 5 (strongly agree) was used.

Residential intention was measured using the question, "What do you think about continuing to live in your current community?" Ordinal scale options were provided with 1 ("I want to keep living here"), 2 ("Maybe I will continue living here"), 3 ("I don't know"), 4 ("I would like to move to another area in the future"), and 5 ("I am already thinking about moving").

A multiple-choice question about why they made the aforementioned choice for the respondents who chose options "I want to keep living here" and "Maybe I will continue living here" was given next; 11 reasons were provided as options in this question. Respondents were asked to select the three options that best matched their thoughts and indicate them in order of importance. Reasons for continuing to live in the current place were related to environmental factors (e.g., "because I like the scenery here"), cognitive factors (e.g., "because I am used to living here"), psychological factors (e.g., "because I am

attached to this place"), social network factors (e.g., "because I have friends or relatives here"), and economic and constraining factors (e.g., "to inherit land or property"). Given the high proportion of those aged ≥65 years in shrinking communities, reasons relating to physical limitations were also included in the options (e.g., "I have difficulty moving because of a physical disability"), etc.

In addition, individual attributes such as age, gender, occupation, residential duration, family composition, vehicle ownership, and housing status were measured using multiple-choice or fill-in questions.

### 3.4. Data Collection and Analysis

The questionnaire survey was conducted in thirty-nine shrinking communities located in eight cities and municipalities in September 2018. Since shrinking communities are usually scattered in remote geographic environments and often contain vacant houses where it is hard to identity whether they are in use, we commissioned the Japan Post Office to deliver questionnaires to valid households in each community in the designated zip code areas. Responses were collected by asking respondents to send back the answered questionnaire.

A total of 1123 questionnaires were mailed via Japan Post to households, and 383 responses were returned to us by mail. 355 of the responses were valid, indicating an overall valid response rate of 31.6%. The mean age of respondents with valid responses was 67.3 years (max = 95, min = 23). Of these respondents, 63.7% (209 people) were men and 36.0% (118 people) were women.

First, the characteristics of residents between two types of shrinking communities were identified. Then, the reasons and motivations of residents for staying were aggregated for each community type, ordered by importance, and compared by using the Chi-square test. Second, environmental factors were extracted using factor analysis and further compared between two types of shrinking communities using *t*-test. Finally, structural equation modeling was conducted based on Figure 2 in order to statistically capture the influence of each dimensional factor on settlement intentions. Meanwhile, the strength of environmental factors on forming RDM factors of each dimension was explored in each type of shrinking community. The discussion was based on the comparison of the self-selected reasons in each community and the structure of RDM, as well as the comparison of the difference in RDM model between two types of communities.

## 4. Results

### 4.1. Preliminary Analysis

#### 4.1.1. Residents in Shrinking Communities

Table 2 displays the characteristics of responses. The response ratio for housing attributes, motor vehicle use, and family composition reflects a common situation of dwellers in shrinking communities. That is, almost all the respondents live in a self-owned property.

**Table 2.** Display of the Sample Characteristics detached house. Most respondents own a car and can drive. Additionally, a large proportion of the respondents live with their families.

| | Characteristic | Suburban Communities | Mountainous Communities |
|---|---|---|---|
| | **Number of Responses** | **n = 158** | **n = 197** |
| Gender | | | |
| | Male | 61.4% | 65.5% |
| | Female | 38.6% | 33.0% |
| Age | | | |
| | The Ratio of Elderly Population | 63.8% | 71.5% |
| | Average Age | 65.5 | 69.8 |

**Table 2.** *Cont.*

| Characteristic | Suburban Communities | Mountainous Communities |
|---|---|---|
| **Number of Responses** | **n = 158** | **n = 197** |
| Residential Duration (years) | 44.3 | 45.2 |
| Family Composition | | |
|     Single household | 15.3% | 16.1% |
|     Couple household | 31.3% | 40.0% |
|     Two generations household | 38.7% | 30.0% |
|     Three generations household | 10.7% | 12.2% |
|     Others | 4.0% | 3.3% |
| Household Attributes | | |
|     Ratio of ownership | 94.2% | 95.4% |
|     Ratio of detached house | 98.1% | 99.5% |
| Mobility Attributes | | |
|     Private car ownership | 92.3% | 88.6% |
|     Ratio of driving by themselves | 90% | 86.2% |
| Residential Intention | | |
|     Intention of staying in current place | 84.0% | 73.1% |
|     Intention of moving out | 5.1% | 16.7% |
|     Not sure | 10.9% | 10.2% |

Note: Unanswered data were excluded.

To capture overall residential intention in shrinking communities, we consider that those who selected "I want to keep living here" or "Maybe I will continue to live here" indicated a desire to continue living in shrinking communities. Conversely, those who selected "I would like to move to another area in the future" or "I am already thinking about moving" expressed a desire to move away from their communities. The results show that the ratio of residents who show continuing intention to live in their current place is 84.0% in suburban communities and 73.1% in mountainous communities. This suggests that most residents wish to continue living in their current area regardless of whether they are in suburban or mountainous communities. This trend is in line with previous Japanese studies that have reported low intention to move [7,8].

4.1.2. Reasons for Residential Intention in Shrinking Communities

The reasons for intention to stay in the current place were aggregated. These reasons were categorized according to order of importance as indicated by the respondents (first, second, and third importance) in Figures 3 and 4. The percentages shown in the bars refer to the percentage of responses obtained for each reason in the specified order of importance.

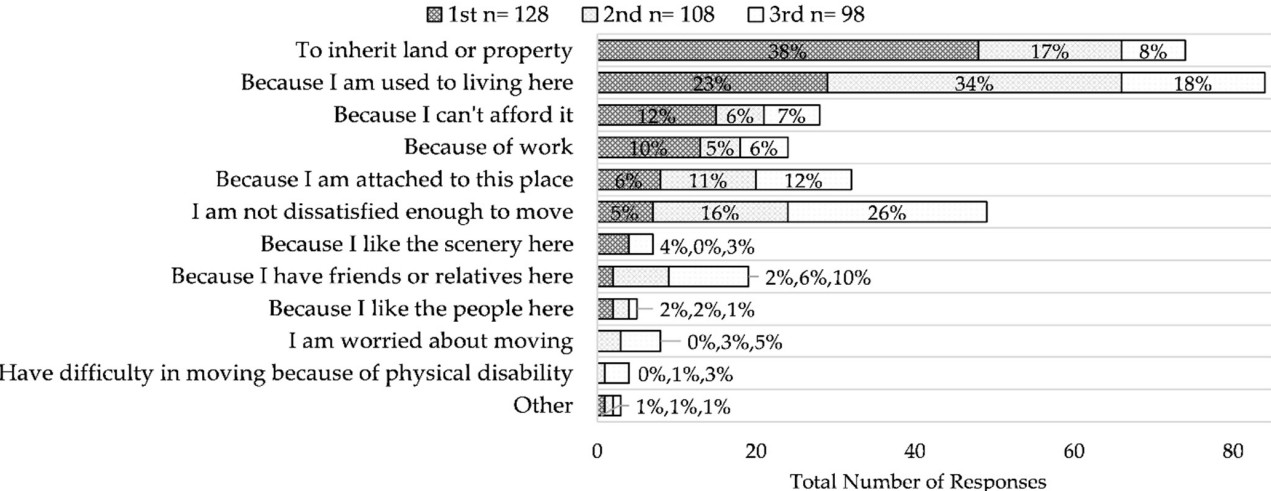

**Figure 3.** Reasons for Continuing Living in Suburban Communities.

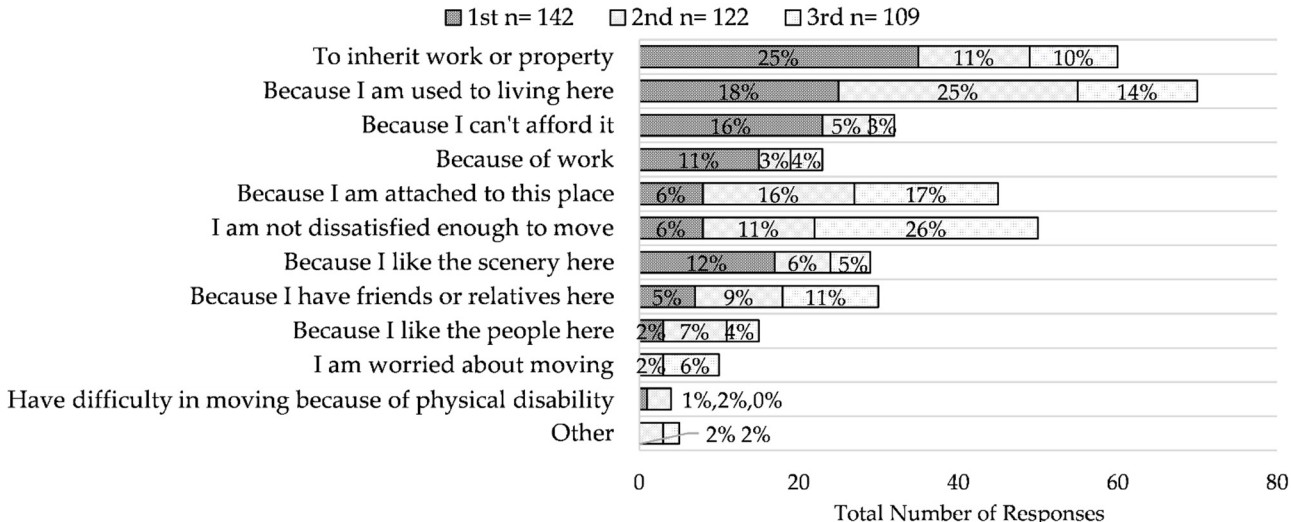

**Figure 4.** Reasons for Continuing Living in Mountainous Communities.

According to the number of responses for each reason (Figures 3 and 4), in both types of communities, being used to living in one's current place was the most common reason given, followed by the inheritance of land and property, lack of strong dissatisfaction with current home, attachment to place, and money constraints. Finally, specific reasons related to work, the local landscape, family, and local bonds were given. The chi-square test examining the relationships between types of shrinking communities and each reported reasons for staying found significance for $\chi^2$ (1, N = 702) = 10.88, $p < 0.001$ **. This result implies that residents in mountainous communities were more likely to view natural environment as a staying reason than suburban communities.

However, when comparing the obtained responses for the most important reason cited, the inheritance of land and property surpasses the other options, followed by habits and money constraints. Place attachment and lack of strong dissatisfaction with the current home were not as important. Further, the chi-square test found that the reason of "To inherit work or property" was more likely to be given as the most important staying reason in suburban communities than mountainous communities, $\chi^2$ (1, N = 270) = 4.64, $p < 0.05$ *). In addition, "Because I like the scenery here" were more likely to be given importance for staying in mountainous communities than in suburban communities, $\chi^2$ (1, N = 270) = 6.16, $p < 0.05$ *).

These results suggest that instead of attachment to the place and convenience of living explored in the prior study [7], a seldom-mentioned motivation for living in a place is the most common reason for residents to continue living there. On the other hand, to inherit land or property shows the strongest binding effect on willingness to settle, which reflects the power of constraints such as ownership on restraining mobility [47,82]. This phenomenon was confirmed in our study in the context of shrinkage. In addition, that inheritance of family property is more valued in suburban areas suggests that a more convenient housing location may strengthen the willingness of residents to inherit property. Furthermore, dwellers in mountainous communities place a higher value on the landscape, suggesting a difference in attractiveness inherent to specific geographical characteristics.

*4.2. Residential Environment in Shrinking Communities*

In order to identify latent environmental elements and attitudes toward shrinking communities, exploratory factor analysis was conducted in the R statistical computing environment. Items concerning the assessment of residential environment and place attachment were used. The KMO test (*MSA* = 0.92) indicated sampling adequacy.

Based on the scree plot and six factors suggested from parallel analysis, a five-factor model was tested using maximum likelihood estimation with Promax rotation. One item

that split across two factors and three items whose factor loadings were below 0.4 in any of the factors were eliminated. The final five-factor model test indicates a good fit (root mean square error of approximation [RMSEA] = 0.065, 90%CI [0.06, 0.07], MRSR = 0.05, comparative fit index [CFI] = 0.954). Table 3 presents the factor loadings.

**Table 3.** Factor Loadings of Exploratory Factor Analysis.

| Items | Components | | | | |
|---|---|---|---|---|---|
| | **1** | **2** | **3** | **4** | **5** |
| Convenient to access central district | **0.79** | 0.01 | −0.11 | 0.04 | 0.04 |
| Convenient to use public transportation | **0.68** | 0 | 0.05 | 0.03 | −0.04 |
| Convenient to commute | **0.67** | 0.07 | −0.12 | 0.06 | 0.07 |
| Convenient to reach medical service | **0.8** | 0 | 0.07 | −0.13 | 0.01 |
| Convenient to go shopping | **0.86** | −0.03 | −0.04 | −0.01 | −0.03 |
| No difficulties in going out | **0.78** | 0.11 | −0.01 | −0.14 | −0.06 |
| Convenient to go to school | **0.84** | −0.08 | −0.09 | 0.05 | 0.07 |
| Convenient to receive welfare service | **0.69** | −0.02 | 0.1 | 0.08 | −0.01 |
| Availability of buses and trains | **0.71** | 0.01 | 0.06 | −0.11 | −0.05 |
| Absence of inconveniences caused by steep slope around home | **0.61** | −0.02 | −0.07 | 0.14 | −0.01 |
| No worries about natural disasters | **0.51** | 0.05 | −0.01 | −0.02 | 0.08 |
| Absence of inconveniences of going on a trip | **0.82** | −0.02 | 0.03 | 0.02 | −0.01 |
| Absence of barriers in using transportation | **0.77** | −0.05 | 0.16 | 0.01 | −0.11 |
| Feeling attached to this place | −0.01 | **0.94** | −0.02 | −0.11 | 0.04 |
| To me, this place is important | −0.02 | **0.93** | −0.09 | −0.02 | −0.01 |
| Feeling happier to live here rather than any other place | 0.08 | **0.8** | −0.07 | 0 | −0.04 |
| Feeling the place is a part of me | 0.05 | **0.8** | 0.02 | −0.06 | −0.04 |
| This place has many important memories for me | −0.04 | **0.58** | 0.11 | 0.13 | 0.11 |
| The place is very special for me | −0.04 | **0.73** | 0.03 | 0.15 | 0.01 |
| Feeling attached to the life here | −0.02 | **0.86** | −0.02 | 0.04 | 0.01 |
| I like this place | 0.04 | **0.86** | 0.03 | −0.01 | −0.06 |
| Someone in neighbor will help you when you are in trouble | −0.01 | −0.07 | **0.98** | −0.04 | −0.02 |
| You can trust your neighbor when you need help | −0.06 | −0.02 | **0.96** | −0.06 | 0.01 |
| You will help the people in trouble as much as possible | 0.04 | −0.06 | **0.58** | 0.1 | −0.04 |
| People here are trustworthy | 0.03 | 0.27 | **0.54** | −0.1 | 0.03 |
| People here should help each other | −0.02 | 0.05 | **0.5** | 0.05 | 0.08 |
| Local scenery is beautiful | −0.17 | 0.17 | 0.02 | **0.54** | −0.1 |
| Local culture should be protected | −0.01 | −0.08 | −0.02 | **0.85** | 0.04 |
| Local history is attractive | 0.08 | −0.04 | 0 | **0.86** | 0.01 |
| Local events are attractive | 0.19 | 0.02 | 0.04 | **0.68** | 0.03 |
| Frequency of participation in local government activities | 0.03 | −0.06 | 0.06 | −0.01 | **0.96** |
| Frequency of participation in local events | −0.05 | 0.04 | −0.01 | −0.01 | **0.83** |

Notes. 1. Residential Convenience, 2. Place Attachment, 3. Social Capital, 4. Attractiveness of Local Asset, 5. Local Participation; Factor loadings >0.40 were bolded.

Factor 1 included thirteen items that measured "residential convenience" with questions regarding factors such as "convenient to use public transportation". Factor 2 included eight items that were assumed to measure "place attachment". Factor 3 included four items that measured "social capital". Factor 4 included four items that measured "attractiveness of local asset". Factor 5 included two items that measured "local participation" with questions about frequency of participation in local government activities or local events.

Table 4 displays the descriptive statistics and correlation coefficients for these five factors, as well as residential satisfaction and cultural norms. The results of the correlations show that three RDM factors are positively related to each other. Both physical and social environment factors were found to be significantly correlated with each RDM factor. Thus, our assumption with respect to RDM structure, namely that environmental factors mediate the RDM factors influencing residential intention, is preliminarily supported.

**Table 4.** Descriptive Statistics and Correlations for RDM factors and Environmental factors.

| Vriables | 1 | 2 | 3 | 4 | 5 | 6 | 7 |
|---|---|---|---|---|---|---|---|
| RDM Factors | | | | | | | |
| 1 Residential Satisfaction | - | | | | | | |
| 2 Place Attachment | 0.71 *** | - | | | | | |
| 3 Cultural Norm | 0.28 *** | 0.40 *** | - | | | | |
| Environmental Factors | | | | | | | |
| 4 Residential Convenience | 0.45 *** | 0.28 *** | 0.28 *** | - | | | |
| 5 Attractiveness of Local Assets | 0.33 *** | 0.50 *** | 0.30 *** | 0.29 *** | - | | |
| 6 Social Cohesion | 0.36 *** | 0.54 *** | 0.39 *** | 0.17 ** | 0.36 *** | - | |
| 7 Local Participant | 0.07 | 0.28 *** | 0.27 *** | 0.02 | 0.15 ** | 0.38 *** | - |
| $\alpha$ | 0.89 | 0.94 | 0.8 | 0.94 | 0.83 | 0.86 | 0.89 |
| $M$ | 3.35 | 3.62 | 3.32 | 2.72 | 3.26 | 3.64 | 3.28 |
| $SD$ | 0.95 | 0.84 | 0.75 | 0.85 | 0.81 | 0.66 | 1.18 |

Notes: n = 355, $p < 0.01$ **, $p < 0.001$ ***.

### 4.3. RDM Structure in Different Types of Shrinking Communities

#### 4.3.1. RDM Factors and Environmental Factors between Shrinking Communities

A *t*-test was applied to find the differences between RDM factors and environmental factors between communities. The results show no significant difference for three RDM factors (*residential satisfaction*, *place attachment*, and *cultural norm*) and for two environmental factors (*local assets* and *local participation*) across communities. In contrast, *residential convenience* ($t$ (1,353) = $-8.58$, $p < 0.001$ ***) was found to be significantly higher in suburban communities than in mountainous communities. *Social capital* ($t$ (1,353) = 2.02, $p < 0.05$ *), on the other hand, was significantly higher in mountainous communities than in suburban communities.

Figure 5 shows the average assessed values of RDM factors and environmental factors in two types of shrinking communities. Because a 5-Point Likert Scale was adopted to measure these factors, it can be assumed that an assessed average above 3 is biased toward a positive evaluation and below 3 toward a negative evaluation. Accordingly, it can be deemed that although these communities are undergoing shrinkage, the RDM factors and most of the environmental factors were rated positively. Only *residential convenience* was assessed negatively in the mountainous communities. This is in line with the cluster analysis results, which indicate that mountainous communities are located in regions further from the nearest train stations and central government offices than suburban communities (Table 1).

#### 4.3.2. RDM Structure between Shrinking Communities

Structural equation modeling was separately applied to verify the RDM structure in two types of shrinking communities in R with the *lavaan* package [83]. In our model, RDM is assumed to be directly predicted by *residential satisfaction*, *place attachment*, and *cultural norms*. Environmental factors indirectly affect residential intention via three RDM factors. In addition, associations between RDM factors, as well as between environmental factors, are also assumed. To determine the fit of the model, root mean square error of approximation (RMSEA), comparative fit index (CFI), and goodness-of-fit index (GFI) are

used [84]. Figures 6 and 7 show the model of RDM structure in suburban and mountainous communities. The validated model was named model 1.

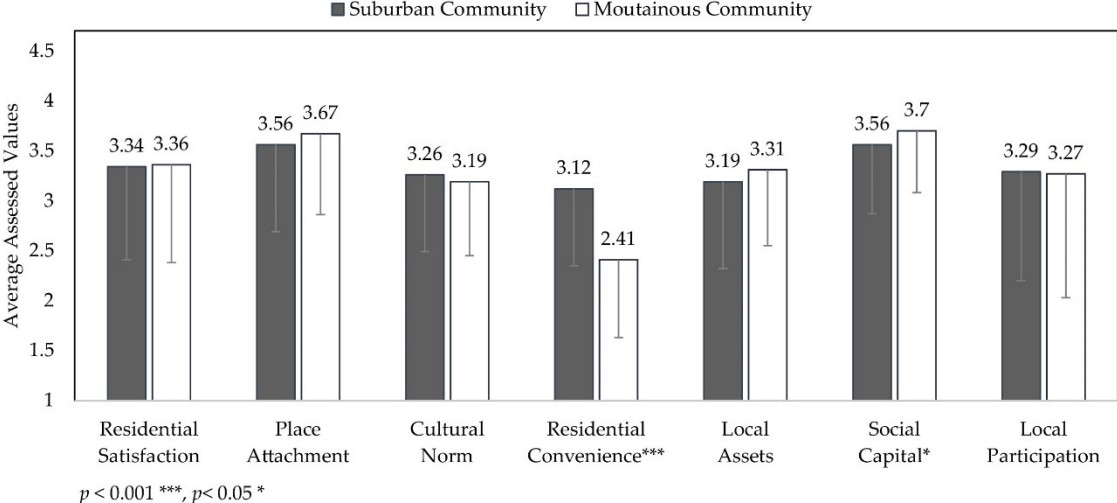

**Figure 5.** Average Assessment of RDM Predictor and Environmental Factors in Two Types of Shrinking Communities.

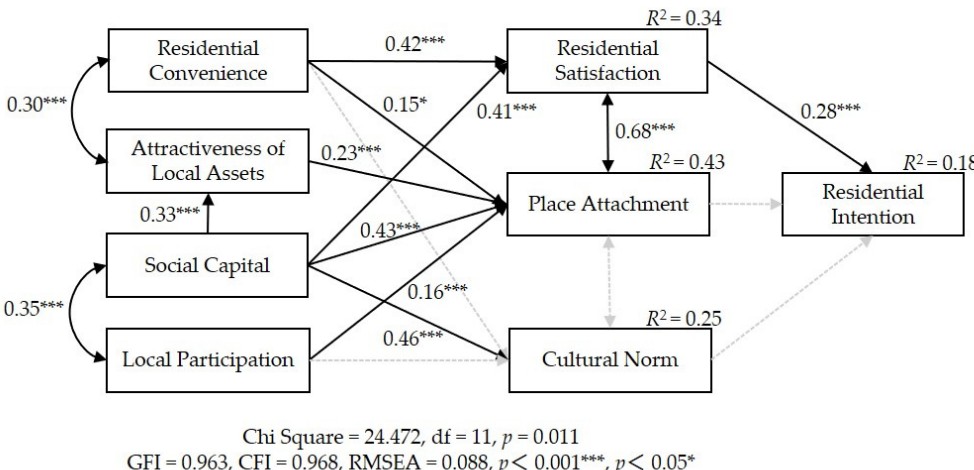

**Figure 6.** RDM Model in Suburban Communities.

The model of suburban communities in Figure 6 shows a good level of fit (GFI = 0.932; CFI = 0.942; RMSEA = 0.094). This model reflects the structure of RDM. That is, residential intention is predicted directly by RDM factor, and the RDM factor is formed by environmental factors. Specifically, residential intention in suburban communities is mainly predicted by *residential satisfaction* ($\beta = 0.28$, $p < 0.001$ ***). This result supports Hypothesis 1 and is in line with previous research results, indicating that low residential satisfaction is strongly related to moving intention for residents of suburban communities [47]. Additionally, no significant direct effect of *place attachment* ($\beta = 0.12$, $p = 0.27$) or *cultural norm* ($\beta = 0.10$, $p = 0.19$) on residential intention was found for these communities, which is not consistent with previous notions [4,5,46]. However, place attachment is strongly correlated with residential satisfaction. Thus, it can be said that psychological utility such as place attachment contributes to residential intention vie the subjective evaluation in suburban communities. Furthermore, in contrast to the result which shows that inheritance of land and property is the most important reason for staying in shrinking communities, a significant effect of *cultural norm* on *residential intention* was not found. Thus, it can be presumed that an obligation in terms of succeeding to a home may not be motivated by normative thinking.

**Figure 7.** RDM Model in Mountainous Communities.

In addition, residential satisfaction can be predicted by both *residential convenience* ($\beta = 0.42$, $p < 0.001$) and *social capital* ($\beta = 0.41$, $p < 0.001$ ***) [5,8]. Meanwhile, place attachment can be predicted by all environmental factors: *residential convenience* ($\beta = 0.15$, $p < 0.05$ *), *attractiveness of local assets* ($\beta = 0.23$, $p < 0.001$ ***), *social capital* ($\beta = 0.43$, $p < 0.001$ ***), and *local participation* ($\beta = 0.16$, $p < 0.001$ ***). Additionally, *cultural norm* is only predicted by *social capital* ($\beta = 0.46$, $p < 0.001$ ***), without any contribution from physical factors.

The model of mountainous communities in Figure 7 also shows a good level of fit (GFI = 0.972; CFI = 0.971; RMSEA = 0.074) and reflects the structure of RDM. Unlike in suburban communities, residential intention in mountainous communities is only predicted by *place attachment* ($\beta = 0.30$, $p < 0.001$ ***) and not by *residential satisfaction* ($\beta = 0.08$, $p = 0.344$) or *cultural norm* ($\beta = 0.08$, $p = 0.245$). Thus, hypothesis 2 is supported. These results furthermore support hypothesis 4, in which the importance of psychological utility would come into play when residential environments are relatively unfavorable.

In comparing the influence of environmental factors on the formation of each RDM factor between communities, differing effects were found. First, although the formation of residential satisfaction in mountainous communities can be predicted by *residential convenience* ($\beta = 0.47$, $p < 0.001$ ***) and *social capital* ($\beta = 0.16$, $p < 0.05$ *), the contribution of *social capital* to *residential satisfaction* shows up differently between communities. A further comparison of model 1 between suburban and mountainous communities using multiple group structural equation modeling was conducted. The model fit with partial constraints on the path of *social capital* on *residential satisfaction* between communities shows significant differences with the configural invariance model ($\Delta$Chi-square = 4.73, $\Delta$df = 1, $p = 0.03$ *). Thus, it can be said that the contribution of *social capital* on *residential satisfaction* appears weaker in mountainous communities than in suburban communities. This decreased emphasis on social environmental factors in mountainous communities is difficult to explain because *social capital* is rated higher in mountainous communities than in suburban communities (Figure 5). This requires further discussion. In addition, *place attachment* is positively predicted by environmental factors such as *residential convenience* ($\beta = 0.18$, $p < 0.01$ **), *attractiveness of local assets* ($\beta = 0.25$, $p < 0.001$ ***), *social capital* ($\beta = 0.33$, $p < 0.001$ ***), and *local participation* ($\beta = 0.11$, $p < 0.05$ *), which is the same composition as the structure of suburban communities. However, *cultural norms* in mountainous communities differ from those in suburban communities, consisting of both *residential convenience* ($\beta = 0.29$, $p < 0.001$ ***) and *local participation* ($\beta = 0.25$, $p < 0.001$ ***) instead of *social capital* ($\beta = 0.12$, $p = 0.078$). These results reflect the possibility that local participation, and housing with desirable accessibility play a greater role in RDM in mountainous communities.

## 5. Discussion

The results from this study indicate that RDM abides by a structure compatible with our assumptions with respect to both types of shrinking communities. That is, residential intention can be directly predicted by RDM factors of multiple dimensions deriving from the residential environment. However, when taking the geographical features of shrinking communities into account, the effects of RDM factors and environmental factors tend to carry different weights within the structure. This result is consistent with research reporting that reasons for continuing to live in shrinking communities could differ on the basis of regional geographical features [4,8]. To contribute to policy measures of sustainable development, such as stabilizing population and improving QoL in shrinking communities, it is necessary to interpret why the RDM structure differs between shrinking communities, as we discuss below.

### 5.1. Differently-Weighted RDM Factors between Shrinking Communities

By comparing the effects of RDM factors on residential intention, it was revealed that staying intention in suburban communities can be predicted by residential convenience, whereas in mountainous communities, it is more directly related to place attachment. This difference suggests that residents in different types of shrinking communities may be inclined to weigh a particular dimension's RDM factor when considering continuous residence in their current place. A plausible theory for interpreting the shift of value weight is cognitive dissonance. Cognitive dissonance refers to a process of cognitive adaptation induced by psychological inconsistencies [85]. For example, studies related to consumer behavior note that the preference and attitude assigned to a product could be altered after a purchase decision [86]. This cognitive adaptation could also exist within the RDM process after people have been living in a particular place for a notable duration.

Because the evaluation of residential convenience in mountainous communities is observed to be lower, it can be presumed that residential environments in those areas do not meet basic daily consumption and mobility needs. In such unfavorable living conditions, it can be speculated that to reach cognitive equilibrium, residents may shift their underpinning RDM factor from an environmental evaluation dimension to an emotional one. Additionally, the result of emphasizing a single dimension factor in RDM suggests that people might not consciously consider all factors during the RDM process. As a study conducted in such a risky village reported, a psychological satisfaction factor like place attachment was shown to be a sufficient reason for dwellers to stay [47]. Schwenk [87] also notes that the cognitive adoption of reaching equilibrium can also be shown to simplify decision making when a residential environment is not as favorable as a resident would hope. On the basis of these interpretations, it can be assumed that RDM may not be as complex as we initially thought. People might simply focus on the more satisfying RDM factors when deciding whether to stay or move.

### 5.2. The Effect of Environmental Factors on Residential Intention between Shrinking Communities

When comparing the impact of environmental factors on residential satisfaction between communities, a shift in residential preference was found if the current neighborhood could not meet a desired expectation. As an example, although residential convenience in the present study was rated higher as a factor in RDM in suburban communities, it was not observed to have a greater effect on shaping residential satisfaction when compared with mountainous communities. The phenomenon of a dominant environmental factor does not necessarily contribute more to the formation of residential satisfaction, as was found in mountainous communities where the dominant factor was social capital. Conversely, although social capital was rated lower as a factor in RDM in suburban communities, it exerts a greater role in the formation of residential satisfaction than in mountainous communities. Similar results were reported by the study of Lovejoy et al. [88], in which suburban residents do not seem to derive more neighborhood satisfaction from features associated with suburban living, such as quietness or low density, nor do traditional

neighborhood residents derive more neighborhood satisfaction from the proximity to destinations. The case of Flanders, furthermore, found that urban residents are more satisfied with peacefulness and safety, whereas rural residents are more satisfied with proximity of facilities [89]. These shifts in the preference of environmental factors imply that favorable environmental resources may be taken for granted, whereas the abundance of unfavorable environmental resources may instead pose a greater impact on the overall evaluation of the residential environment.

*5.3. Characteristics of RDM in Shrinking Communities*

After understanding the validated RDM structure in conjunction with the reasons for staying as reported by residents, settlement behavior in shrinking communities was found to be strongly influenced by the constraints of property and cognition. Thus far, RDM studies have maintained the viewpoint that RDM is dominated by the actual environment, implying that a particular residential mobility behavior is caused by certain features of an alternative place that can better meet a resident's particular needs [37,55]. However, our results show that neither the absence of dissatisfaction nor attachment to place, which have been considered mainstream RDM factors, are the main reasons for intention of continuous residence. Instead, inheriting land and property, being accustomed to living in the current place, and monetary constraints stood out as the decisive reasons. This further suggests that settlement behavior in shrinking communities tends to be determined by the constraints of property and money, which is caused by the decline of local economic development. Conversely, Coulter et al. [32] noted that the longer a person lives in a particular place, the less likely that person will be to act on a desire to move. Considering the generally long residential duration of residents in shrinking communities (Table 2), it could be assumed that habit could be a nonnegligible cognitive characteristic of residents remaining in shrinking communities. In sum, it can be concluded that settlement behavior in shrinking communities shows a passive characteristic, rather than being maintained by pull factors such as landscape or favorable social climate.

## 6. Conclusions

In this study, a questionnaire survey was conducted to clarify the RDM structure in shrinking Japanese communities. The main results can be summarized in two points. First, staying intention in different types of communities is influenced by different dimensions of the RDM structure. Residents in suburban communities with relatively convenient geographical features tend to give more weight to the overall evaluation of the environment in considering staying, whereas residents in mountainous communities are more likely to give a higher value to RDM factors from the psychological dimension, such as place attachment. Second, regardless of the type of community, severe unfavorable environmental factors will profoundly impact a resident's RDM. Conversely, RDM may not benefit from additional favorable environmental factors. In sum, settlement behavior in Japanese shrinking communities largely displays a passive nature, driven by cognitive and monetary constraints.

Two policy implications can be derived from these results. First, an effective strategy for promoting staying intention in shrinking communities requires capturing the key RDM factors that residents in a particular community value. For example, communities could focus on enhancing place attachment in mountainous areas and prioritizing the fitness of the residential environment in suburban communities. In this regard, the importance of promoting social capital should be given attention. Furthermore, effective housing management policies and monetary support would directly contribute to intention of settlement. Second, environmental design thinking could represent a larger contribution to subjective residential satisfaction than functional development in shrinking communities. Our results imply that residents of Japanese shrinking communities may experience cognitive adaptation to their place of residence. To enrich residential experience and perception, local governments should establish platforms for promoting local participation,

advancing community development strategies, and beautifying neighborhoods through artistic creation.

Finally, because the survey in this study lacked a detailed measurement of the objective environment of each respondent, it is difficult to further expand upon the interactions between residents and their objective environments. For future research, taking socioeconomic variables into consideration could be an important entry point for understanding residents' perceptions of their communities and for promoting community sustainability.

**Author Contributions:** Conceptualization, S.A., T.A. and A.S.; methodology, T.A. and A.S.; software, S.A. and T.A.; validation, S.A. and T.A.; formal analysis, S.A. and T.A.; investigation, S.A. and T.A.; resources, S.A., T.A. and A.S.; data curation, S.A.; writing—original draft preparation, S.A.; writing—review and editing, S.A., T.A. and A.S.; visualization, S.A.; supervision, T.A.; project administration, T.A. and A.S.; funding acquisition, T.A. and A.S. All authors have read and agreed to the published version of the manuscript.

**Funding:** This work was supported by KAKENHI, Grant-in-Aid for Scientific Research (B:21H01449) and (C:18K04382).

**Institutional Review Board Statement:** The study was conducted according to the guidelines of the Declaration of Helsinki, and approved by the Ethics Committee of Graduate School of International cultural Studies, Tohoku University, Japan (Approval Code: No.6, 7 December 2018).

**Informed Consent Statement:** Informed consent was obtained from all respondents involved in the study.

**Data Availability Statement:** Because the primary data collected in this study is based on a commitment to disclose the results of statistical analysis to obtain resident responses, the furthermore data present are available on the request from the corresponding author.

**Acknowledgments:** The authors would like to acknowledge the supported of KAKENHI in providing opportunities that enabled this work and the assistance of Japan Post in conducting the survey. The authors would like to thank Enago (www.enago.jp (accessed on 5 November 2021)) for the English language review.

**Conflicts of Interest:** The authors declare no conflict of interest.

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
