# Peer review of "Why People Settle in Shrinking Communities: A Unified Framework of Psychological, Environmental, and Cognitive Factors"

_sustainability, doi:10.3390/su132413944_

Round 1

Reviewer 1 Report

  • A definition of shrinking communities is needed in the introduction with a justification of the selection of that particular definition. Perhaps a global definition of the concept then a localized one for Japan. 
  • Consider revising the title and shortening it while taking into consideration the keywords. 

Author Response

I am very grateful for your comments on the manuscript. We amended the relevant part of the manuscript according to your advice.

Point 1: A definition of shrinking communities is needed in the introduction with a justification of the selection of that particular definition. Perhaps a global definition of the concept than a localized one for Japan. 

Response 1:This suggestion is very helpful for us to emphasize the functional difference in conducting surveys on the neighbourhood scale and city scale.

We added the discussion about the definition of the shrinking city in the 3rd paragraph, and the definition of the shrinking community in the 4th paragraph of the Introduction in the manuscript.

Point 2: Consider revising the title and shortening it while taking into consideration the keywords. 

Response 2: We shortened the title as follows:

Why People Settle in Shrinking Communities: Unified Framework of Psychological, Environmental, and Cognitive Factors

 We would like to thank you for your time to review our manuscript.

Reviewer 2 Report

This manuscript presents a well documented study and one that will be useful for future research and policy. I suggest the following minor comments for the manuscript. In the introduction, there should be an introduction of quality of life and previous research, especially those related with the topic on hand. The header of section 3.4 would be more appropriately called 'Data collection' or 'Procedure' as conduction is not an accurate term here. Throughout the manuscript, do ensure all figures are consistently presented in 2dp and with the leading 0 for figures less than 1.

Author Response

I am very grateful for your comments on the manuscript. These suggestions are helpful for us to better organize the factors related to RDM investigated in previous studies. It also allows us to better clarify the originality of our study. We amended the relevant parts in the manuscript according to your advice as follows.

 Point 1: In the introduction, there should be an introduction of quality of life and previous research, especially those related to the topic on hand.

Response 1: An introduction of quality of life and previous research was provided in the Introduction. We listed them based on the types of factors and then presented the common findings regarding their effect on Residential Decision Making. The revised discussions are shown in the 2nd paragraph of the manuscript.

Point 2: The header of section 3.4 would be more appropriately called 'Data collection' or 'Procedure' as conduction is not an accurate term here.

Response 2: “Survey Conduction” was renamed based on your advice to “Data Collection and Analysis Procedure”

Point 3: Throughout the manuscript, do ensure all figures are consistently presented in 2dp and with the leading 0 for figures less than 1.

Response 3: All figures have been standardized to 2-digit precision with leading 0, where applicable.

We would like to thank you for your time to review our manuscript.

Reviewer 3 Report

The paper ‘Why People Settle in Shrinking Communities: A Framework of Residential Decision Making Considering Psychological, Environmental, and Cognitive Dimensions’ submitted to Sustainability aims to make a relationship between several predictor that can have an effect on residential decision making. Based on a survey of (n=) 355 authors conclude for a relationship between psychological, environmental and cognitive dimensions for residential decision making. The paper is well develop both in theory as well as methodologically. Data treatment is finely developed. There are minor points that authors could consider: (i) research design should be better discussed since it compromises the ‘puzzle’ (Blaikie and Priest 2019) you are trying to solve; (ii) a more detail account for data treatment should be given, especially when correlations and chi-square employment do not give you and account of which predictors are stronger. Good luck with your research!

Author Response

I am very grateful for your comments on the manuscript. These suggestions are helpful for us to learn more about research design and deepen our understanding of our research method. We also better clear our research design during the revisions. We amended the relevant parts in the manuscript according to your advice as follows.

Point 1: research design should be better discussed since it compromises the ‘puzzle’ (Blaikie and Priest 2019) you are trying to solve.

Response 1:

Thank you so much for your recommendation of the book "Designing Social Research, The Logic of Anticipation". Based on your comments and the four tasks (Focusing task, Framing task, Selecting task and Distilling task) comprising the design's choices and specification mentioned in the book, we improved our research design by declaring: 

  • Research question
  • Appropriate reasons for determining our research method were noted based on a review of methods in previous studies.
  • The type of research method with a detailed description of data collection was declared
  • The data collection and analysis procedure was improved by detailing the source of data, sampling.

The revisions are detailed in Research Design and Method in the manuscript.

Point 2: a more detail account for data treatment should be given, especially when correlations and chi-square employment do not give you and account of which predictors are stronger.

Response 2:

We are sorry for the unclear description of the analysis procedure.

We answered your comments based on the understanding of the following question are asked.  

1)how to compare the weight of different staying reasons within the cluster.

2)How to compare the effect of predictors on residential intention using SEM.

For 1) We measure the importance of staying reasons by providing ordered selection question( the 1st important reason; 2nd important reasons; 3rd important reasons) for respondents, which is used to compare the difference among these reasons directly.

For 2) We mainly focus on comparing on differences on significant path in structural modelling analysis between communities. We assumed the significant predictors indicate that factor is important for the residents living in the corresponding types of community.

Based on your comments, we summarized our data analysis procedure at the end of Research Design and Method in the manuscript, which may provide the whole picture for the whole analysis flow. 

We would like to thank you for your time to review our manuscript.